# Combined XPO1 Inhibition and Parthenolide Treatment Can Be Efficacious in Treating Triple-Negative Breast Cancer

**DOI:** 10.3390/ijms262010243

**Published:** 2025-10-21

**Authors:** Amy L. Paulson, Radwa M. Elmorsi, Adam M. Lee, R. Stephanie Huang

**Affiliations:** 1Department of Molecular Pharmacology and Therapeutics, School of Medicine, University of Minnesota, Minneapolis, MN 55455, USA; pauls860@umn.edu; 2Department of Experimental and Clinical Pharmacology, College of Pharmacy, University of Minnesota, Minneapolis, MN 55455, USA; radwa.elmorsy@med.tanta.edu.eg (R.M.E.); leeam@umn.edu (A.M.L.); 3Department of Pharmacology, Faculty of Medicine, Tanta University, Tanta 31527, Egypt

**Keywords:** triple negative breast cancer (TNBC), selective inhibitors of nuclear export (SINEs), XPO1, parthenolide, NFKBIA, combination therapy

## Abstract

Triple-negative breast cancer (TNBC) is an aggressive, heterogeneous subtype of breast cancer with limited treatment options. Our previous work explored repurposing selinexor, an XPO1 inhibitor, as a novel therapeutic option for TNBC. To enhance its efficacy, this study aimed to identify beneficial combination therapies with selinexor and experimentally evaluate their effects in TNBC. Using the computational tool IDACombo, we nominated drugs predicted to improve the efficacy of XPO1 inhibition. The top candidate, parthenolide, was tested in vitro using three transcriptionally distinct TNBC cell lines. Fluorescently labeled cells were co-cultured and treated with selinexor, parthenolide, or their combination. Growth inhibition was assessed across the mixed population and by individual cell line after 96 h, and potential synergy was evaluated using Combenefit. While selinexor and parthenolide monotherapy inhibited the growth of TNBC subtypes, the combination was more effective in suppressing the overall cell population. Synergistic interactions between the two agents were observed in specific TNBC lines but not all, reflecting the combination effect in heterogeneous TNBC patients. Our findings suggest the selinexor–parthenolide combination as a potential therapeutic strategy for TNBC, warranting further investigation. Our study also demonstrates the value of integrative computational–experimental approaches in guiding heterogeneity-informed drug combinations for preclinical evaluation.

## 1. Introduction

Breast cancer remains the most commonly diagnosed cancer among women in 2025, accounting for approximately 32% of all new cases [1,2]. Among its various subtypes, triple-negative breast cancer (TNBC) is recognized as a highly aggressive and clinically challenging form, making up about 15–20% of all breast cancer diagnoses in the U.S. [3]. TNBC is defined by the lack of estrogen receptor (ER), progesterone receptor (PR), and human epidermal growth factor 2 (HER2) expression, leaving patients with very limited targeted therapeutic options [4]. Moreover, TNBC is the most heterogeneous breast cancer subtype, exhibiting diverse molecular and pathological characteristics that complicate treatment and hinder the predictability of therapeutic responses [5]. Due to this heterogeneity, coupled with its propensity for the rapid development of resistance to monotherapy, combination therapies have been proposed as a strategy to improve clinical outcomes [3]. Combination approaches are hypothesized to enhance overall treatment efficacy by targeting distinct subpopulations within a tumor, each potentially responsive to a different agent within the combination, thereby improving clinical outcomes [6].

Selinexor, a selective inhibitor of exportin 1 (XPO1), is a U.S. Food and Drug Administration (FDA)-approved medication for the treatment of multiple myeloma [7,8]. Furthermore, XPO1 has been reported to be highly expressed in various tumors including breast cancer, making it a promising therapeutic target [9]. In our previous work, we identified and validated selinexor as a potential therapeutic agent for TNBC through a computational drug repurposing approach. By imputing drug sensitivity scores from two independent datasets of breast cancer patient tumors, we identified agents for which TNBC patients were predicted to have heightened sensitivity compared to other breast cancer subtypes. Selinexor emerged as a top candidate, and we subsequently validated its efficacy in a panel of TNBC cell lines. Our analyses further revealed a strong association between selinexor sensitivity in TNBC and the inhibition of the NF-κB signaling pathway [10]. Despite these promising findings, the great heterogeneity of TNBC and its tendency to rapidly develop resistance to monotherapies highlight the need for combination strategies to maximize the therapeutic potential of selinexor [3].

To rationally identify effective drug combinations, we utilized IDACombo, a computational tool that has been shown to accurately predicts efficacy of drug combinations based on the independent drug action (IDA) theory [11]. Specifically, IDACombo was developed through the analysis of high-throughput monotherapy screening data in cancer cell lines, enabling the identification of promising therapeutic combinations using IDA as a reference principle. Predictions generated by IDACombo have demonstrated a strong correlation with in vitro measured combination efficacy (Pearson’s r = 0.93 for over 5000 drug combinations) and achieved >84% accuracy in predicting clinically meaningful improvements across 26 first-line therapy trials [11]. This underscores its robustness and utility as a predictive tool for rational drug combination design in cancer therapy. Moreover, IDACombo offers concentration-dependent predictions, enabling the estimation of combination efficacy at clinically relevant drug concentrations [11]. IDA posits that the efficacy of a drug combination in a cell line or patient is expected to be equivalent to that of the most effective single agent within the combination. The higher the IDACombo score generated, the higher the combination efficacy in a heterogeneous cancer setting with no requirement that each component of the combination interacts with each other [6]. This approach is particularly well-suited to heterogeneous diseases such as TNBC, which exhibits substantial inter- and intra-tumoral heterogeneity, resulting in diverse molecular subtypes and highly variable treatment responses both across and within patients [12]. In this heterogeneous landscape, IDACombo provides a practical advantage over traditional synergy-based models, as it identifies combinations where each drug may act independently on different subpopulations within a tumor or patient population. For example, one drug in the combination may be effective in one molecular subtype, while the second drug targets another, thereby expanding the overall proportion of responsive patients [11]. In contrast, traditional synergy models typically assume biological uniformity and require interaction between agents to produce greater-than-expected efficacy, an assumption that may not hold true in a disease as heterogeneous as TNBC [6]. Therefore, IDACombo provides a more reliable and broadly applicable approach for identifying effective drug combinations across heterogeneous tumor environments. In the present study, we aimed to identify a rational combination therapy with selinexor using IDACombo and validate it in TNBC cell lines in the hope to enhance selinexor’s therapeutic efficacy and expand its clinical utility in the treatment of TNBC.

## 2. Results

### 2.1. Parthenolide Is Predicted to Be Highly Efficacious in Combination with an XPO1 Inhibitor

TNBC is characterized by high intertumoral heterogeneity, which presents a major challenge for identifying broadly effective therapies. To address this, we employed IDACombo, which models drug combinations under the principle of IDA, prioritizing combinations that maintain efficacy across diverse cell states and tumor profiles rather than relying on synergy. Screening across breast cancer cell lines from the CTRPv2 dataset (*n* = 40), we evaluated all compounds (*n* = 481) in combination with leptomycin B, an XPO1 inhibitor. Parthenolide emerged as the top candidate, achieving the highest IDACombo score among all screened combinations (Figure 1A). Additionally, parthenolide appeared as the 6th and 15th most predicted efficacious combination with KPT-185 and compound 7d-cis, respectively. These are the two other XPO1 inhibitors from the CTRPv2, our initial screening dataset.

To further assess its potential, we visualized the predicted and observed responses across all concentration pairs in three-dimensional space for leptomycin and parthenolide (Figure 1B). Each data point represents a specific dose combination, with the gray plane indicating the maximal effect achievable by either monotherapy alone. Notably, many of the predicted parthenolide–leptomycin B combinations (pink dots) lie below this plane, supporting their joint potential to enhance therapeutic efficacy beyond single-agent treatment. This suggests that parthenolide may broaden the response profile of XPO1 inhibitors and increase its clinical utility across a heterogeneous TNBC population.

### 2.2. Hierarchical Clustering Captures the Heterogeneous Landscape of TNBC Patient Tumors and Enable Selection of Representative TNBC Cell Lines

To contextualize our IDACombo predictions within a more clinically relevant framework, we first sought to assess how well our TNBC cell line models reflect the diversity of real patient tumors. We integrated batch-corrected gene expression data from 27 TNBC cell lines and 326 basal-like breast cancer samples from the METABRIC cohort. PAM50 basal status was used as a proxy for TNBC, as clinical receptor status was not available for all patients and basal-like tumors most closely resemble TNBC at the molecular level. We then performed hierarchical clustering and dimensionality reduction to assess the structure of intertumoral heterogeneity across both datasets.

Hierarchical clustering using Ward’s linkage revealed three major transcriptional clusters that spanned both patient tumors and cell lines (Figure 2A). Notably, TNBC cell lines did not segregate into a single artificial cluster but instead distributed across the full range of patient subtypes. The top cluster (green) was composed of nine cell lines: MDA-MB-468, BT-20, HCC-1143, HCC-1187, HCC-1599, HCC-1937, HCC-2157, HCC-70, and HCC-38. The middle cluster (orange) contained ten cell lines, including HCC-1806, CAL-148, DU-4475, MFM-223, SUM-185PE, SUM-229PE, CAL-851, SUM-102PT, SUM-149PT, and SUM-159PT. The bottom cluster (blue) consisted of the remaining eight cell lines: MDA-MB-231, MDA-MB-436, BT-549, CAL-120, CAL-51, HCC-1395, HS-578T, and MDA-MB-157. Only 16 of the 326 patient tumor samples were not clustered with a cell line (gray).

Further analysis of the transcriptional profiles of the clustered cell lines revealed that these groupings were consistent with known molecular subtypes of TNBC. TNBC subclassification have been broken down into basal-like (BL1 and BL2), an immunomodulatory (IM), a mesenchymal (M), a mesenchymal stem-like (MSL), and a luminal androgen receptor (LAR) subtype [12,13]. The MDA-MB-231 cluster was largely composed of highly mesenchymal and invasive cell lines, reflecting its known characteristics. The MDA-MB-468 cluster consisted of cell lines with basal-like characteristics, while the HCC-1806 cluster contained a mixture of cell lines known to be highly heterogeneous, with characteristics from mesenchymal, LAR, and other heterogeneous subtypes. This concordance between our clustering results and known TNBC subtypes validates our approach for selecting representative cell lines to model a broad range of tumor heterogeneity.

The three cell lines selected for our functional studies—MDA-MB-468, HCC-1806, and MDA-MB-231—were representatives from these distinct clusters. This approach enabled us to map in vitro model systems onto a patient-relevant expression landscape, ensuring that combination therapies nominated through IDACombo—such as parthenolide + an XPO1 inhibitor—have the potential to address the molecular diversity present across real patient TNBC tumors.

To representatively model this heterogeneity, we used these cell lines and engineered them to stably express distinct fluorescent proteins (GFP, RFP, and BFP, respectively). These lines were co-cultured and monitored for proliferation over 96 h in drug-free conditions to assess baseline cellular growth. As shown in the growth rate comparison in Figure 2B, all three fluorescently labeled cell lines maintained a stable exponential growth rate when co-cultured. Statistical analysis revealed no significant difference in the calculated growth rate constant (k) for MDA-MB-231, MDA-MB-468, or HCC-1806 when grown in co-culture compared to their respective parental lines in monoculture (*p* > 0.05). This confirmed that fluorescent tagging did not alter growth dynamics and that the co-culture system provides a stable platform for parallel evaluation of drug responses across transcriptionally distinct TNBC subtypes.

Together, these results validate the representativeness and transcriptional diversity of the TNBC cell lines used as proxies for patient tumors and support the use of this co-culture framework for functionally testing therapeutic combinations across diverse molecular backgrounds.

### 2.3. The Combination of Parthenolide and Selinexor Enhances Cell Growth Inhibition in Mixed TNBC Cell Lines Model

To investigate the efficacy of the combination treatment in inhibiting heterogenous TNBC cell growth, we evaluated the growth inhibition effects of parthenolide and selinexor alone or together across the co-culture mixture of TNBC cell lines. The mixed population was treated with parthenolide (1, 3, 6, 9 µM), selinexor (10, 25, 50, 100, 150, 200 nM), or their combination. Drug concentrations were selected based on prior dose–response screening experiments, focusing on IC_50_ values and concentrations spanning the IC_10_–IC_90_ range for each agent. Among the tested doses, parthenolide at 6 µM and selinexor at 50 nM demonstrated the most pronounced inhibitory effects on overall population growth while avoiding excessive cytotoxicity. Fluorescent images were acquired, and signals were quantified in each channel to evaluate treatment effects on individual cell lines, with total fluorescence measured to determine the overall impact of treatment on the mixed population.

As shown in Figure 3A, both selinexor and parthenolide exhibit significant single-agent cytotoxicity when evaluated across all three cell lines combined (total cell population), reducing viability to 54.11 ± 5.62% and 68.08 ± 7.01% of control, respectively (*p* < 0.001). Notably, combination treatment revealed substantially enhanced cytotoxicity, further reducing viability to 23.62 ± 4.15%, significantly lower than either monotherapy (*p* < 0.001). In Figure 3B, the fluorescent images of each treatment condition visually reflect the trends observed in the percent viability data. In the control group, a dense and balanced distribution of all three TNBC cell lines was observed. Treatment with parthenolide alone resulted in a moderate reduction in overall cell density compared to control, with more pronounced effect observed in MDA-MB-468 and HCC-1806 cell lines. Selinexor treatment also led to a noticeable decrease in total cell population, with the reduction appearing more uniform across all three cell lines. Importantly, the combination treatment exhibited the lowest density of fluorescently labeled cells, indicating the highest level of cytotoxicity across all cells. To further evaluate the effect of each compound within the mixed-cell model, we analyzed the response of the individual cell lines. This approach was based on the fact that TNBC is a highly heterogeneous disease, where patients may exhibit distinct molecular subtypes with varying treatment sensitivities. Accordingly, our co-culture model was designed using three TNBC cell lines, each representing a unique molecular subtype, to mimic this intertumoral heterogeneity. Given the expected variability in drug response among these subtypes, we compared cellular viability across the three TNBC cell lines following each treatment condition.

In MDA-MB-231 cells (Figure 3C), parthenolide alone did not significantly reduce viability compared to control (107.6 ± 19.63%), suggesting that MDA-MB-231 is relatively unaffected by parthenolide. In contrast, selinexor significantly reduced viability to 29.52 ± 5.87% (*p* < 0.001), indicating sensitivity to this agent. Combination treatment resulted in 25.4 ± 3.64% viability, which is comparable to selinexor monotherapy, revealing that the addition of parthenolide did not enhance cytotoxicity in this cell line model. However, the combination was significantly more effective than parthenolide alone (25.4 ± 3.64% vs. 107.6 ± 19.63% viability; *p* < 0.001), confirming the dominant cytotoxic role of selinexor in this line. These findings are supported by the images shown in Figure 3B, where the parthenolide-treated group displays a higher abundance of BFP-labeled cells, indicating less sensitivity of MDA-MB-231 to parthenolide. In contrast, the selinexor and combination treatment groups show fewer BFP-labeled cells, suggesting a greater response of MDA-MB-231 to these treatments.

MDA-MB-468 cells (Figure 3D) exhibited moderate sensitivity to both parthenolide (63.4 ± 7.40%, *p* < 0.001) and selinexor (70.7 ± 6.50%, *p* < 0.001) as single agents. The combination treatment significantly enhanced cytotoxicity, reducing viability to 32.33 ± 6.59% (*p* < 0.001 vs. either agent alone), demonstrating that the combination treatment exerted greater cytotoxicity. No significant difference between the two single agents, indicating comparable potency at the tested doses.

HCC-1806 cells (Figure 3E) demonstrated the most pronounced cytotoxic response. While both parthenolide and selinexor showed moderate single-agent activity (43.55 ± 14.47% and 64.72 ± 7.54% viability, *p* < 0.001 and *p* < 0.01, respectively), their combination dramatically reduced viability to 15.27 ± 6.17%, significantly lower than either agent alone (*p* < 0.001 versus parthenolide, *p* < 0.05 versus selinexor), alluding to a possible enhanced cytotoxic effect in this cell line. The effects of selinexor and parthenolide as single agents were comparable, with no significant difference between the two groups. Figure 3B confirms the findings shown in Figure 3D,E, with fewer GFP- and RFP-labeled cells observed in the parthenolide and selinexor treatment groups compared to the control.

Taken together, the combination of selinexor and parthenolide enhanced the cytotoxicity compared to either compound alone in the total cell population, and in two of the three individual cell lines (MDA-MB-468 and HCC-1806). MDA-MB-231 cells appeared to be unaffected by parthenolide but remained highly sensitive to selinexor.

### 2.4. Combenefit Synergy Analysis Confirms Cell Line-Specific Interactions Between Parthenolide and Selinexor

To assess the potential synergistic interactions between parthenolide and selinexor on each of the TNBC cell lines, we treated MDA-MB-231, MDA-MB-468, and HCC-1806 cells separately with increasing concentrations of parthenolide, selinexor, and their combination. Cell viability was evaluated after 96 h by measuring cell area normalized to the control. Data from three biological replicates was analyzed using Combenefit (version 2.02), applying the Highest Single Agent (HSA) model to quantify drug interactions. The results were visualized as 3D surface synergy maps and heatmap matrices of synergy scores.

In MDA-MB-231 cells (Figure 4A,D), the 3D surface plot revealed some regions of blue shading, indicating localized synergistic interactions at specific concentration of the combinations. However, the corresponding heatmap matrix did not show any statistically significant synergy scores across the tested range.

MDA-MB-468 cells (Figure 4B,E) displayed broader areas of blue regions in the 3D surface plot compared to MDA-MB-231, suggesting greater potential synergy. The heatmap matrix identified statistically significant synergistic interactions at two concentrations of the combinations, supporting a modest but meaningful enhancement of drug efficacy in this cell line.

HCC-1806 demonstrated the most robust synergistic interaction of the drug combinations. The 3D surface plot showed extensive blue regions with higher intensity and broader area, indicating strong synergy across multiple dose combinations. The corresponding heatmap matrix confirmed significantly elevated synergy scores at several concentrations, highlighting a consistent and potent synergistic effect. These findings align with earlier observations from the mixed-cell model, in which HCC-1806 exhibited the greatest reduction in viability in response to the combination treatment.

In summary, Combenfit synergy analysis using the HSA model revealed cell line-specific responses to parthenolide and selinexor combination. While MDA-MB-231 exhibited minimal interaction, MDA-MB-468 showed moderate synergy, and HCC-1806 demonstrated strong synergistic cytotoxicity.

## 3. Discussion

TNBC remains the most aggressive subtype of breast cancer, defined by the absence of ER, PR, and lack of HER2 amplification [4]. The lack of molecular targets limits treatment options and contributes to the poor prognosis associated with TNBC [3]. Additionally, TNBC is further complicated by expressing extensive inter- and intratumoral heterogeneity, In turn driving substantial treatment response variability across and within patients [5]. In our previous work, we identified and validated the potential efficacy of selinexor, an FDA-approved XPO1 inhibitor, as an effective monotherapy for TNBC treatment [10]. However, the pronounced heterogeneity in response and the rapid emergence of resistance to monotherapy highlights the clinical necessity of combination strategies to potentiate selinexor efficacy and prevent resistance [14]. Combination therapies simultaneously target multiple pathways or distinct subpopulations within a tumor offering a promising strategy to overcome this heterogeneity and improve clinical outcomes [6].

In the current study, we employed a previously validated computational tool, IDACombo, to predict potential compounds that can be combined with an XPO1 inhibitor to enhance its therapeutic efficacy [11]. IDACombo is based on the independent drug action (IDA) model which allows each drug acting independently on different tumor subpopulations to achieve populational level efficacy. The core principle of IDA is to maximize the proportion of patients within a population who respond to at least one drug, thereby increasing the overall number of individuals who achieve a therapeutic response [6]. This approach is particularly advantageous for heterogeneous diseases like TNBC because, unlike traditional synergy models that assume biological uniformity and rely on direct interaction between agents, it can target a wider range of molecular subtypes, thereby expanding the overall proportion of responsive patients. Leveraging this framework, IDACombo nominated the natural product parthenolide as a promising candidate for combination with the XPO1 inhibitor selinexor, and this prediction was experimentally validated in a heterogeneous TNBC co-culture model. Our findings showed that the selinexor–parthenolide combination enhanced cytotoxicity beyond either monotherapy in the heterogeneous TNBC cohort; meantime, synergistic combination effect may also exist in selected TNBC subtypes.

IDACombo identified parthenolide as the top candidate to be combined with XPO1 inhibitor in breast cancer from a total of 481 agents screened. The high IDACombo score indicates that, even when considering the inherent cellular diversity of TNBC, parthenolide consistently broadens the efficacy profile of XPO1 inhibition. Furthermore, the strong ranking of parthenolide in combination with other XPO1 inhibitors, namely KPT-185 and compound 7d-cis, further supports the robustness of this interaction across structurally distinct XPO1 inhibitors. Three-dimensional dose–response visualizations provide additional evidence, demonstrating that many parthenolide-XPO1 inhibitor dose combinations exceed the maximal effect achieved by monotherapy, indicating superior efficacy beyond that of the individual agents. These results, coupled with those of our selinexor combination screening, suggest a possible class-wide effect between XPO1 inhibitors and parthenolide.

Parthenolide is a sesquiterpene lactone derived from the medicinal plant feverfew (*Tanacetum parthenium*), long recognized for its diverse pharmacological properties, including anti-inflammatory, anti-cancer, and anti-migraine effects [15,16,17,18]. In breast cancer, parthenolide has demonstrated antiproliferative and pro-apoptotic effects in various cell lines, including TNBC models [19,20,21]. Its anti-cancer activity is largely attributed to inhibiting the NF-κB signaling pathway [22]. NF-κB is a crucial transcription factor that regulates the expression of genes involved in cell proliferation, survival, inflammation, and angiogenesis. Furthermore, its constitutive activation is frequently observed in various cancers, including breast cancer [21,23]. By suppressing NF-κB activity, parthenolide can inhibit tumor growth, induce apoptosis, and sensitize cancer cells to conventional chemotherapies [24]. The safety profile of parthenolide, primarily evaluated through feverfew-based preparations in early-phase clinical trials, suggests good tolerability with no dose-limiting toxicities at tested doses, despite challenges due to limited systemic bioavailability [25]. Furthermore, parthenolide selectively induces apoptosis in tumor cells while sparing normal cells, suggesting a favorable therapeutic index. It also protects some normal cells from oxidative stress damage, indicating low intrinsic toxicity to healthy tissue [15].

The inhibition of XPO1 has gained significant attention as a promising therapeutic avenue in TNBC, supported by multiple studies demonstrating that XPO1 blockade effectively diminishes cell viability and induces apoptosis across diverse TNBC models [10,26,27]. Our previous work demonstrated that selective inhibition of XPO1 with selinexor robustly suppresses TNBC cell viability and induces apoptosis across diverse TNBC cell lines, with IC_50_ values well below clinically achievable plasma concentrations [10]. Importantly, our data indicated that sensitivity to XPO1 inhibition does not correlate directly with XPO1 expression levels. Instead, this cytotoxic effect is primarily mediated through inhibition of the NF-κB pathway, achieved by nuclear retention of NFKBIA (IκBα), a key negative regulator of NF-κB signaling. Under physiological conditions, XPO1 facilitates the nuclear export of cargo proteins such as NFKBIA into the cytoplasm. In contrast, selinexor treatment leads to the nuclear accumulation of NFKBIA, where it sequesters NF-κB, thereby preventing its nuclear translocation and subsequent transcription of survival and proliferation genes. This mechanism was substantiated in our study by knockdown of *NFKBIA*, which resulted in resistance to selinexor, underscoring its essential role in mediating sensitivity to XPO1 inhibition and suggesting that therapeutic efficacy of selinexor depends on the functional reliance on the nuclear export of critical cargo proteins like NFKBIA. Our transcriptomic analyses further revealed significant modulation of both canonical and non-canonical NF-κB pathway genes following selinexor treatment, reinforcing the centrality of NF-κB signaling in the response to XPO1 inhibition [10]. Building on these findings, the computational prediction by IDAcombo that parthenolide, a well-characterized NF-κB inhibitor, is a promising agent to combine with selinexor aligns with our mechanistic data. Parthenolide-mediated NF-κB inhibition is expected to enhance selinexor sensitivity by intensifying NF-κB pathway blockade, thereby broadening the responsive TNBC cell population and augmenting therapeutic efficacy. This provides a strong mechanistic rationale for the combination, moving beyond computational prediction to a biologically supported strategy.

To test the broad efficacy of parthenolide and selinexor combination in heterogeneous TNBC patient population, we intentionally used an unbiased transcriptomic similarity method to select TNBC cell lines that may broadly represented the heterogeneous TNBC patient tumors. We integrated gene expression data from TNBC cell lines and 326 basal-like patient tumors (METABRIC cohort) revealing three major transcriptional clusters encompassing both clinical and cell line samples. Importantly, cell lines did not form isolated artificial clusters but clustered intermixed with patient tumors, consistent with prior molecular profiling studies demonstrating TNBC division into multiple biologically and clinically relevant molecular subtypes [12,28]. Our clustering recapitulated known TNBC subtypes: MDA-MB-231 cells aligned with mesenchymal/invasive phenotypes; MDA-MB-468 with basal-like subtypes; and HCC-1806 represented a mixed subtype cluster including luminal androgen receptor characteristics. This validates the representativeness of our cell line models for patient tumors, justifying their use in experimental screening. By employing a co-culture model composed of three molecularly distinct TNBC cell lines, we rigorously represented the transcriptomic intertumoral heterogeneity characteristic of TNBC in patients. Additionally, we observed diverse treatment sensitivities typical of clinical patient treatment response and were able to dissect both population-wide and subtype-specific responses to treatment. This multiplexed approach strengthens the translational relevance of our results by ensuring that observed effects are not restricted to a single subtype but are broadly applicable across diverse molecular backgrounds. Moreover, the fluorescent labeling and co-culture of these representative lines enabled simultaneous assessment of drug responses across molecularly distinct TNBC subtypes within a shared environment. Consistent growth kinetics between co-cultured and single lines verified the biological fidelity of this system, which enhances translational relevance of our co-culture model.

When selinexor and parthenolide were used as single agents in our co-culture model, both agents reduced overall cellular viability, with selinexor showing stronger sensitivity compared to parthenolide (dosages used were in the nM range compared to µM). Consistent with our findings, selinexor has shown broad single-agent antitumor activity in TNBC models both in vitro and in vivo, including patient-derived xenografts, making it a clinically promising agent [26,27,29]. Parthenolide has also been extensively investigated for its anti-cancer properties across a range of malignancies, including breast cancer [19,21]. We demonstrate that combination treatment significantly enhanced cytotoxicity compared with either agent alone; however, analysis at the individual cell line level revealed distinct response patterns. MDA-MB-231 cells showed relative resistance to parthenolide monotherapy at the tested dose, but remained highly sensitive to selinexor. The addition of parthenolide did not further enhance cytotoxicity in this cell line at the tested concentration. This illustrates the critical importance of using experimental models that adequately capture tumor heterogeneity when evaluating combination therapies. Without accounting for this diversity, there is a risk of drawing inaccurate conclusions about the true effectiveness of a combination. Nonetheless, this cell line demonstrated the greatest sensitivity to selinexor, supporting the concept that combination treatment can increase the likelihood of achieving positive outcomes in a heterogeneous population, even when some subtypes exhibit resistance to one component. In contrast, the combination therapy resulted in marked reductions in viability for both MDA-MB-468 and HCC-1806, indicating greater sensitivity to the combination compared to monotherapies. This highlights its ability to effectively target molecularly diverse TNBC subtypes that might otherwise respond differently to single agents. Notably, the most dramatic reduction was observed in HCC-1806, suggesting that specific molecular backgrounds may be particularly susceptible to the combination.

Combenefit is a user-friendly tool for visualizing, analyzing, and quantifying drug combination effects in terms of synergy or antagonism. It supports classical synergy models, including Loewe Additivity (Loewe), Bliss Independence (Bliss), and Highest Single Agent (HSA), and can process data from single experiments or high-throughput screens. Although our computational prediction is grounded in the IDA principle, the possibility of synergism between the two agents is not contradictory but rather reflects a distinct and complementary mechanism contributing to combination therapy efficacy. IDA posits that the therapeutic benefit of a combination arises primarily from inter-patient heterogeneity, wherein different patients respond preferentially to different agents, thereby increasing the likelihood that any given patient benefits from at least one drug in the regimen [6]. In contrast, synergism refers to a supra-additive interaction in which the combined effect of the drugs exceeds the expected additive effect of each agent alone [30]. Importantly, these two concepts are not mutually exclusive: IDA emphasizes independent activity across heterogeneous patient populations, whereas synergism highlights cooperative interactions at the cellular or molecular level. Thus, even though our initial combination nomination was primarily guided by the IDA concept, it remains valuable to evaluate synergistic interactions at the level of individual cell lines. Moreover, we investigated the potential for additional synergy as the drug combination identified through IDA is known to act mechanistically on the same pathway. Since each cell line represents a distinct TNBC subtype, such analyses can reveal subtype-specific synergism that may contribute to enhanced therapeutic outcomes. Combenefit synergy analysis provided valuable insights into the cell line-specific nature of interactions between parthenolide and selinexor in TNBC [31]. Among the three synergy models available within the software, we selected the Highest Single Agent (HSA) model. This model assesses the drug combination effect by comparing it solely to the most effective individual agent, without considering the expected additive effects of both drugs combined. This approach is conceptually aligned with the IDA principle previously used to predict combination efficacy in patient cohorts [30]. By systematically testing a matrix of drug combinations and quantifying interactions using HSA model, our analysis reveals that the efficacy of this drug pair is highly dependent on the molecular context of individual TNBC cell lines. In MDA-MB-231 cells, the analysis indicated only limited and localized regions of synergy, with no statistically significant synergy scores observed across most concentration ranges. MDA-MB-468 cells exhibited broader synergy across concentration ranges, with statistically significant interactions found at select dose pairs. This moderate, yet meaningful, synergistic effect underscores the importance of dosing strategy and molecular context in optimizing combination regimens. The most robust synergy was observed in HCC-1806 cells, where both the 3D synergy maps and heatmaps highlighted extensive and statistically significant synergy. This cell line, which harbors distinct basal-like features and exhibits high molecular heterogeneity, responded most profoundly to dual pathway inhibition.

While our in silico nomination and in vitro experimental validation suggest that the selinexor-parthenolide combination holds promise as a treatment strategy for TNBC, it is crucial to acknowledge the inherent limitations of the current study. A primary limitation is the exclusive reliance on in vitro models, which do not fully recapitulate the complex in vivo tumor microenvironment. These models inherently lack the diverse cellular interactions, signaling molecules, and extracellular matrix components present in patient tumors. Therefore, additional in vivo studies are essential to further advance this combination towards clinical adaptation. Furthermore, future research is needed to optimize dosing strategies and to determine if comparable therapeutic effects can be achieved at clinically relevant and tolerable concentrations. Despite these limitations, strong in vitro evidence provides a compelling rationale for continued development of this combination.

## 4. Materials and Methods

### 4.1. Data Acquisition

The Molecular Taxonomy of Breast Cancer International Consortium (METABRIC) study and associated clinical annotations were downloaded directly using the MetaGxBreast Bioconductor package within r [32]. The Broad Institute’s Cancer Cell Line Encyclopedia (CCLE) TPM-normalized RNA expression values and corresponding cell line annotations were obtained using the Depmap r package (version 1.22.0) [33]. Prediction models were constructed from combining The Cancer Therapeutics Response Portal Version 2 (CTRPv2) [34] cancer cell line drug response database with the cancer cell line gene expression data from the Broad Institute’s CCLE.

### 4.2. Constructing Models and Predicting Combination Drug Scores for TNBC Patients

To computationally estimate the efficacy of drug combinations across breast cancer subtypes, we applied the IDACombo framework to the CTRPv2 dataset. Gene expression profiles were log-transformed and standardized prior to analysis. For each CTRPv2 breast cancer sample, predicted AUC values for each individual drug combination and dosage were computed via the IDACombo R package (version 1.28.0) [11,35].

To estimate the viability of drug combinations, we employed the IDA model, which assumes the effect of a drug combination is driven by the most effective constituent agent. Therefore, we can predict combination efficacy by selecting the maximal response each cell line has to multiple drugs in a given combination. This is then repeated for each cell line and the average is taken to create a mean drug effect on a population. We compare the predicted mean combination effect to the observed mean monotherapy effect and generate the IDACombo score. Statistical analysis was performed by conducting 10,000 Monte Carlo simulations of the combination effect using the variance in drug response data. Specifically, we evaluated the predicted combination viability of XPO1 inhibitors contained within the CTRPv2 (*n* = 3) with all other drugs in the dataset (*n* = 481) and applied it across all breast cancer samples (*n* = 40). For each drug pair, the predicted IDA combination response was calculated as the minimum predicted viability (or maximum predicted AUC) between the two drugs, reflecting the expected effect under the assumption of independent drug action. Higher IDACombo scores are indicative of higher predicted combination efficacy. The resulting IDACombo scores provide a conservative estimate of combination efficacy, without assuming synergy.

All analyses were performed using R version 4.5.1 with IDACombo version 1.0.2 or the dedicated IDACombo web shiny app.

### 4.3. Development of Heterogenous Mixed Cell Line Model for TNBC

#### 4.3.1. Representative TNBC Cell Line Selection

Transcriptomic profiles from breast cancer patients and cell lines were obtained from the METABRIC cohort and CCLE, respectively. METABRIC microarray expression data were filtered using “best_probe” annotations to retain the most representative probe per gene. Common genes between both datasets were identified and retained for downstream analysis. Log-transformed expression values were used throughout and genes with low variance (bottom 25%) across samples in each dataset were excluded. After filtering, expression matrices from CCLE and METABRIC were concatenated by shared genes.

To mitigate systematic differences between the CCLE and METABRIC datasets, batch correction was performed using ComBat from the sva r package [36]. Corrected expression values were used for all subsequent analyses. Principal Component Analysis (PCA) was used to visualize the data before and after batch correction. Additionally, density plots of expression distributions by batch and silhouette scores by dataset were generated. Following batch correction, expression data were subsetted to include only TNBC breast cancer cell lines and basal-like patient tumor. Cancer cell lines were identified using TNBC subtype metadata and METABRIC basal tumor samples were classified based on PAM50 status.

Hierarchical clustering was performed using the Ward.D2 method on Euclidean distances calculated from the transposed batch-corrected matrix. To evaluate clustering stability and optimality, silhouette width scores were computed. Cluster number (k) was selected based on maximized average silhouette width and visual inspection. Dendrograms were generated using the dendextend package [37].

All analyses were performed using R version 4.5.1 with sva version 3.56.0 and dendextend version 1.19.1.

#### 4.3.2. Cell Maintenance and Culture

Three human TNBC cell lines: HCC-1806 (RRID:CVCL_1258), MDA-MB-231 (RRID:CVCL_0062), and MDA-MB-468 (RRID:CVCL_0419) were purchased from the American Type Culture Collection (ATCC, Manassas, VA, USA). Cells were cultured in phenol red-free RPMI-1640 media supplemented with 10% FBS at 37^◦^C and 5% CO2 atmosphere. All cell lines were routinely tested and confirmed negative for mycoplasma contamination using the Universal Mycoplasma Detection Kit protocol (ATCC, Manassas, VA, USA).

#### 4.3.3. Lentivirus-Cells Transduction

Fluorescently labeled cells were generated using lentivirus transduction with a distinct fluorescent color for each cell line: MDA-MB-231 (BFP), MDA-MB-468 (GFP), and HCC-1806 (RFP). Successfully transduced cells were selected using puromycin and lentivirus used for transduction was purchased from VectorBuilder Inc. (Chicago, IL, USA).

#### 4.3.4. Co-Culture Growth Assay and Drug Treatment

MDA-MB-231, MDA-MB-468, and HCC-1806 cell lines were trypsinized, harvested, and resuspended in phenol red-free RPMI 1640 medium supplemented with 10% FBS. The resuspended cells were combined to a final concentration of 5 × 10^4^ cells/mL and seeded into 96-well plates (Thermo Scientific, Sunnyvale, CA, USA) at a density of 5 × 10^3^ cells per well. Ratios between cells within the mixture were adjusted according to the growth rate of each cell line to achieve nearly equal representation, ensuring that the results were not driven by a single cell line. Cells were allowed to attach for 24 h. Following incubation, cells were treated with selinexor (10, 25, 50, 100, 150, or 200 nM), parthenolide (1, 3, 6, or 9 μM), the drug combination (pairwise concentrations), or vehicle control (media).

### 4.4. Drug Preparation

Selinexor and parthenolide were obtained from Selleck Chemicals (Houston, TX, USA), reconstituted in DMSO, and aliquoted into 10 mM stocks stored at −80 °C. The medium was used to dilute selinexor and parthenolide to its final concentration and a DMSO of no greater than 0.1% by volume was used for all experiments.

### 4.5. Cellular Viability Assessment in the Mix Cell Model

Percent viability was measured 96 h post-treatment by quantifying the change in relative fluorescent cell area (96 h/0 h) via live-cell imaging. The resulting data were normalized to no-drug controls for each experimental plate and final calculations performed in GraphPad Prism (v10.5.0).

Cell Imaging was performed every 24 h using the Cytation™ Cell Imaging Multi-Mode Reader (Gen5 software, version 3.14), (BioTek Instruments, Winooski, VT, USA) in the BFP, GFP and RFP channels. Automatic background flattening parameters were used to remove the background fluorescence from the BFP, GFP, and RFP channels and object masking thresholds were set to identify each cell for measuring area. All results are reported as a mean and standard deviation of 3 independent biological replicates, each containing a minimum of 3 technical replicates for each treatment condition.

### 4.6. Synergy Analysis Using Combenefit

Synergistic interactions between selinexor and parthenolide were analyzed using Combenefit software (version 2.02; Cancer Research UK Cambridge Institute). Cell lines were trypsinized, harvested, and resuspended in phenol red-free RPMI 1640 medium supplemented with 10% FBS to a final concentration of 5 × 10^4^ cells/mL, then seeded into 96-well plates (Thermo Scientific, Sunnyvale, CA, USA) at a density of 5 × 10^3^ cells per well. Each cell line was plated individually and allowed to attach for 24 h before treatment. Selinexor was tested at concentrations ranging from 5 to 20 nM, and parthenolide was tested at concentrations ranging from 35 to 800 μM, to generate individual dose–response curves for each drug and each cell line. Cells were also treated with combinations of these concentrations or with vehicle control (media). Normalized values from single-agents and combination treatments were imported into Combenefit for analysis. Synergy was assessed using the Highest Single Agent (HSA) reference model. Combenefit generated three-dimensional surface plots and heatmap matrices, along with calculated summary synergy scores based on the HSA model [31]. The software is freely available and was downloaded from SourceForge.

### 4.7. Ethics

No identifiable human data were included in this study.

### 4.8. Role of Funders

This study was supported by NIH/NCI Grants R01CA204856 (R.S.H) and a University of Minnesota (UMN) Office of Academic Clinical Affairs (OACA) Grant-in-Aid Program (GIA) award. R.S.H. also received support from NIH/NCI R01CA229618, Advanced Research Projects agency for Health (ARPA-H) ADAPT program award, UMN OACA Faculty Research Development grant, a UMN Masonic Cancer Center CRTI Exceptional Translational Research award, UMN College of Pharmacy SURRGE award and UMN IPUC-the Prostate and Urologic Cancer TWG pilot award. The funders did not influence the research discovery made in this study.

## 5. Conclusions

In conclusion, this in vitro study provides a strong rationale for combining XPO1 inhibition with parthenolide to address the molecular diversity and resistance to monotherapy that characterize TNBC. By integrating computational prediction with experimental validation in a heterogeneous model that reflects clinically relevant diversity, we show that parthenolide enhances the therapeutic potential of selinexor and expands its activity across molecularly distinct TNBC subtypes. This combination further demonstrated cell line-specific synergistic interactions. More broadly, our findings emphasize the value of heterogeneity-informed experimental models for predicting clinical responses and guiding the rational design of targeted combination therapies in TNBC and other heterogeneous malignancies. Overall, this combination holds promise for overcoming the therapeutic challenges of TNBC.

## Figures and Tables

**Figure 1 ijms-26-10243-f001:**
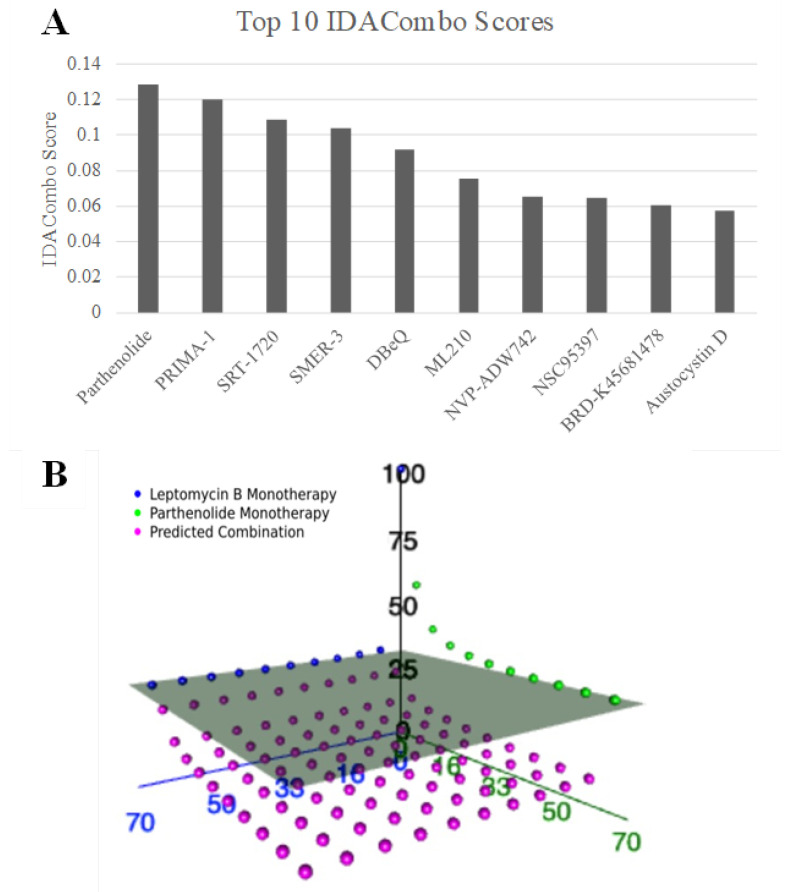
Parthenolide with an XPO1 inhibitor is predicted to be the most efficacious drug combination for breast cancer. (**A**) Bar plot showing the top 10 IDACombo scores predicted for combinations of leptomycin B (an XPO1 inhibitor) with all other drugs screened in the CTRPv2 dataset (*n* = 481) using breast cancer cell lines (*n* = 40). Higher IDACombo score is indicative of higher predicted combination efficacy. (**B**) 3D plot illustrating measured and predicted average cellular viabilities across 27 breast cancer cell lines. The blue and green spheres represent the viability for leptomycin B and parthenolide monotherapies, respectively, at various concentrations. The pink spheres represent the predicted viability of the combination treatment. The gray plane represents the lowest average viability achievable with either monotherapy alone; therefore, pink spheres that fall below this plane indicate that the combination therapy is predicted to reduce cellular viability more effectively than either monotherapy alone.

**Figure 2 ijms-26-10243-f002:**
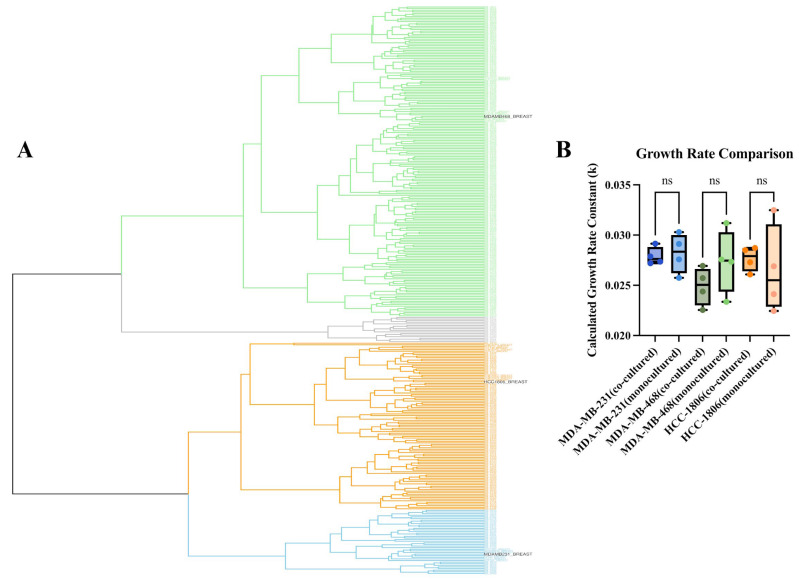
Hierarchical clustering of TNBC patient tumor samples and cell lines captures meaningful subpopulations for representing intertumoral heterogeneity. (**A**) Batch-corrected hierarchical clustering of gene expression data from 326 basal-like patient tumors (METABRIC cohort) and 27 CCLE TNBC cell lines. Representative cell lines selected from each cluster for further functional study are highlighted in black. (**B**) Box plot showing the calculated growth rate constant (k) for fluorescently labeled TNBC cell lines (MDA-MB-231-BFP, MDA-MB-468-GFP, HCC-1806-RFP) grown in co-culture compared to their respective parental lines in monoculture. Fluorescently labeled cells grown in co-culture proliferated exponentially over a 96 h period in the absence of drug exposure, showing no significant difference in their calculated growth rate compared to their parent lines. Statistical analysis was performed using an unpaired *t*-test: ns, not significant.

**Figure 3 ijms-26-10243-f003:**
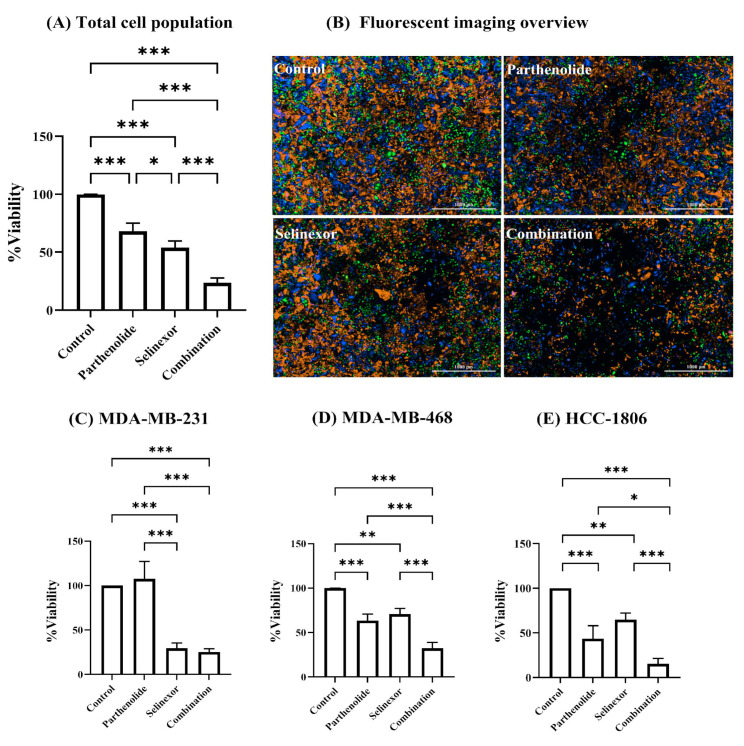
Combination of parthenolide and selinexor exhibit enhanced cellular growth inhibition in a mixed TNBC co-culture model. Three TNBC cell lines were fluorescently labeled using distinct lentiviral vectors: MDA-MB-231 (BFP, blue), MDA-MB-468 (GFP, green), and HCC-1806 (RFP, orange). The cells were mixed in a co-culture model and treated with either media only (Control), parthenolide (6 µM), selinexor (50 nM), or a combination of both agents. Fluorescent images were captured using the Cytation™ Cell Imaging Multi-Mode Reader (Gen5 software, version 3.14). Percent viability was assessed based on normalized fluorescent cell area. (**A**) Bar graph showing percent viability of the total cell population. (**B**) Representative fluorescent images of the co-culture under each treatment condition. (C–E) Bar graphs showing individual viability responses of each cell line within the co-culture: (**C**) MDA-MB-231, (**D**) MDA-MB-468, and (**E**) HCC-1806. Data represents mean ± SD from three biological replicates (*n* = 3). Data and images were collected 96 h post-treatment. Statistical analysis was performed using one-way ANOVA followed by post hoc multiple comparisons. Significance levels: * *p* < 0.05; ** *p* < 0.01; *** *p* < 0.001.

**Figure 4 ijms-26-10243-f004:**
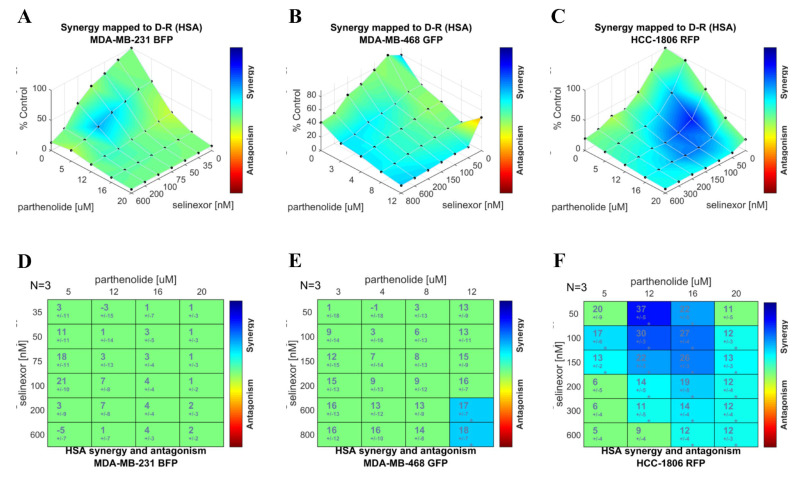
Parthenolide and selinexor combination demonstrates cell line-specific synergistic interactions in TNBC based on Combenefit analysis using Highest Single Agent (HSA) model. Combenefit software (version 2.02) was used to evaluate the combinatorial effects of parthenolide and selinexor on cell viability in three TNBC cell lines independently: MDA-MB-231, MDA-MB-468, and HCC-1806. Cells were treated individually with increasing concentrations of parthenolide, selinexor, and their combination. Cell viability was measured as normalized cell area from three biological replicates (*n* = 3). Top panel (**A**–**C**): 3D surface plots represent drug combination effects in (**A**) MDA-MB-231, (**B**) MDA-MB-468, and (**C**) HCC-1806, mapped according to HSA model. The x- and y-axes represent parthenolide and selinexor concentrations, respectively, and the *z*-axis represents percent cell viability (% Control). Blue regions indicate synergistic interactions while regions leaning towards red indicate possible antagonistic interactions. Bottom panel (**D**–**F**): Corresponding HSA synergy scores are shown as heatmap matrices for (**D**) MDA-MB-231, (**E**) MDA-MB-468, and (**F**) HCC-1806. Numbers in each cell represent the synergy score ± SD from three independent experiments (*n* = 3). Values are color-coded according to the obtained score, with blue indicating statistically significant synergistic interactions. The vertical color scale bar to the right of each plot or matrix reflects the gradient of synergy or antagonism with synergy towards blue color and antagonism towards red color. Statistical significance was assessed using a one-sample *t*-test (Significance levels: * *p* < 0.05).

## Data Availability

All data generated or analyzed during this study are included in the published article.

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
