# Peer review of "Combined XPO1 Inhibition and Parthenolide Treatment Can Be Efficacious in Treating Triple-Negative Breast Cancer"

_ijms, 2025, doi:10.3390/ijms262010243_

Round 1
Reviewer 1 Report
Comments and Suggestions for Authors
The manuscript reports on combined XPO1 inhibition and parthenolide treatment can be efficacious1 in treating triple-negative breast cancer. The manuscript may be publishable but needs a minor revision.
- As for the introduction section, the description of background of the considered topic is too superficial and should be expressed in more detail. For example, before introducing triple-negative breast cancer, the author should briefly introduce some basic knowledge about breast cancer. You can refer to this article for that (Medical Gas Research, 2026 16(1):p 41-45). This will help readers understand the significance of this article.
- There are many spellings that should be noted and revised. For example, “IC50” should be “IC50”, it should be “MDA-MB-231” not “MDAMB231”, etc..
Reviewer 2 Report
Comments and Suggestions for Authors
The article, titled "Combined XPO1 Inhibition and Parthenolide Treatment Can Be Efficacious in Treating Triple-Negative Breast Cancer," presents a computational and experimental approach to optimize combination therapies for triple-negative breast cancer (TNBC). Using the IDACombo (Independent Drug Action) model, the authors identified parthenolide as the most promising compound to enhance the efficacy of the exportin-1 (XPO1) inhibitor selinexor. The combination demonstrated enhanced cytotoxicity in heterogeneous co-culture models of TNBC and synergy in specific cell lines (HCC1806 > MDA-MB-468 > MDA-MB-231). The main conclusion is that combined inhibition of XPO1 and NF-κB (mediated by parthenolide) can expand therapeutic efficacy and overcome the resistance observed in monotherapies, providing a mechanistic and translational basis for the development of personalized combination therapies.
The title is clear and concise, reflecting the focus of the research. However, it could specify that the results are limited to in vitro studies, avoiding premature clinical inference.
The abstract is well-structured, self-contained, and without citations. It adequately summarizes objectives, methods, and conclusions. However, it does not mention limitations or the in vitro nature of the study, which can lead to translational overinterpretation.
The introduction provides a solid contextualization of the heterogeneity of TNBC and justification for the use of combinations. The rationale for IDACombo is well integrated, but there may be a flaw in the reasoning: the text assumes that IDA theory "guarantees" population efficacy, without discussing model limitations or cases of inapplicability (suggesting causality where there is only statistical correlation). In the results (L104-122), the graphs and explanations are clear, but they lack explicit reference to reproducibility (number of replicates, detailed statistical testing). It is recommended to specify the statistical method used to compare IDA predictions vs. observed results.
In Lines 139-187 of the results, the authors demonstrate excellent genomic integration for cellular model selection. However, as a possible methodological limitation, it may be pertinent to note that grouping by gene expression lacks independent validation (e.g., quantitative silhouette score or heatmap of discriminating genes).
In Lines 203–321 of the results, the experimental analysis is robust and visually well documented. However, there appears to be a lack of reasoning in the discussion of the results (lines 263–266 and 305–308): the authors state "enhanced cytotoxicity" and widespread "synergy," but the data show a significant effect in only two cell lines, not all—it should be rephrased as "cell line–dependent enhancement."
In the discussion (L332–492), the wording is clear and logical, and contextualized with the appropriate literature. We suggest rectifying the overstatement (lines 356–358 and 629–633): the conclusions extrapolate to "promising therapeutic strategy in TNBC" without in vivo or pharmacokinetic data. It appears There is still a lack of discussion of methodological limitations, namely: (1) lack of validation in animal models, (2) possible interference from supraphysiological concentrations of parthenolide, (3) exclusive reliance on in silico expression models for selection.
The figures and tables are clear, with descriptive captions and in-text citations. There are no apparent inconsistencies.
The references are up-to-date (2020–2025) and relevant. However, there appear to be some duplication (refs. 11 and 4 are identical). Standardizing the DOI format is recommended.
The conclusions (628–639) are consistent with the results, but should include explicit mention of the limitations and suggest the need for in vivo validation.
This manuscript has appropriate scientific language and is fluently written at the level appropriate for Cancers or Molecular Oncology. Slight reduction of redundancy in the discussion is recommended.
In conclusion, this manuscript is within the scope of molecular oncology journals, but requires adjustments. in the discussion and conclusion to avoid overstatement and ensure alignment with standards of scientific rigor and reproducible transparency.
